# Association of Intraocular Pressure and Optical Coherence Tomography Angiography Parameters in Early Glaucoma Treatment

**DOI:** 10.3390/diagnostics12092174

**Published:** 2022-09-08

**Authors:** Lan-Hsin Chuang, Ju-Hsien Li, Pei-Wei Huang, Henry S. L. Chen, Chun-Fu Liu, Ju-Wen Yang, Chi-Chun Lai

**Affiliations:** 1Department of Ophthalmology, Chang Gung Memorial Hospital, Keelung 204201, Taiwan; 2College of Medicine, Chang Gung University, Taoyuan 333323, Taiwan; 3Department of Ophthalmology, Chang Gung Memorial Hospital, Linkou 333423, Taiwan; 4Program in Molecular Medicine, National Yang Ming Chiao Tung University, Taipei 112304, Taiwan

**Keywords:** glaucoma, capillary vessel density, optical coherence tomography angiography, intraocular pressure, retinal nerve fiber layer

## Abstract

This prospective study aimed to explore the effect of medical intraocular pressure (IOP) reduction on structural and capillary vessel density (VD) change by optical coherence tomography (OCT) angiography in early glaucoma. Patients with newly diagnosed glaucoma and a follow-up of ≥6 months were enrolled. An ocular examination that included slit-lamp bio-microscopy, pneumatic tonometry, gonioscopy, standard automated perimetry, and OCT angiography was performed. Quantitative OCT angiography parameters were assessed using a linear mixed model that was adjusted for inter-eye correlation. The correlations between IOP changes and OCT angiography parameter changes were analyzed using Spearman’s correlation test. In total, 52 eyes of 36 participants, including 33 glaucoma eyes of 17 participants and 19 healthy eyes of 19 participants served as the case and control groups, respectively. The IOP of the case group decreased from a baseline mean of 20.4 ± 0.8 mmHg to 15.7 ± 0.5 mmHg at 3 months (*p* < 0.001) and to 16.1 ± 0.5 mmHg at 6 months (*p* < 0.001). For the subgroup with an IOP reduction of >20%, the deep macula VD was negatively correlated with baseline IOP and significantly decreased at 3 months follow-up. Additionally, change in retinal nerve fiber layer (RNFL) was positively correlated with a change in IOP at 6 months. In conclusion, the deep-layer macula VD was correlated with baseline IOP and influenced by the reduction in IOP in the short term. The changes in VD revealed the vulnerability of the deep vascular complex. The OCTA parameters provide in vivo monitoring information during medical treatment for early glaucoma.

## 1. Introduction

Glaucoma is a group of optic neuropathies that lead to the progressive loss of retinal ganglion cells, thinning of the retinal nerve fiber layer, and degeneration of the optic nerve [1]. The exact pathogenesis of glaucoma remains unclear. However, insufficient vascular supply appears to play a crucial role in the pathogenesis of glaucomatous eyes [2]. The development of optical coherence tomography (OCT) angiography has allowed for qualitative and quantitative evaluations of the microvasculature at the macula, optic nerve head (ONH), and peripapillary region [3,4]. Compared with healthy individuals, reduced vessel density (VD) has been observed in individuals with various stages of glaucoma [3]. Moreover, previous studies also report an association between a decrease in macular VD and severity and progression of visual field (VF) loss [4].

Intraocular pressure (IOP) reduction is generally accepted as the standard treatment for patients with glaucoma [5,6]. Both medical and surgical IOP reduction interventions can slow or even halt the progression of glaucoma [5,6]. However, the findings pertaining to the effect of IOP reduction on microvascular change in patients with glaucoma are inconsistent. Some studies have revealed an improvement in OCT angiography parameters after IOP reduction treatment [7,8,9,10,11], whereas others have not [12,13,14]. Moreover, most studies have focused on the effect of surgical IOP reduction [10,11,13,14] instead of medical IOP reduction [7,8,15], which is the most common initial choice for treating glaucoma. A prospective investigation of the changes in the ocular microvasculature induced by IOP-lowering medications may help to clarify the effect of long-term IOP reduction treatments on glaucoma.

In the present prospective study, the ocular microvasculature of the macula, optic nerve head, and peripapillary region was examined using OCT angiography. The study aimed to determine the effects of medical IOP reduction on OCT angiography in treatment-naïve patients with early glaucoma.

## 2. Material and Methods

### 2.1. Study Population

Patients with newly diagnosed primary angle glaucoma (POAG), normal-tension glaucoma (NTG), or ocular hypertension (OHT) were prospectively enrolled as participants in the present study during the period between April 2020 and July 2021. The unaffected healthy fellow eyes of the enrolled patients were used as controls. All participants were examined at Keelung Chang Gung Memorial Hospital and followed up for at least 6 months; additionally, they all received topical IOP-lowering medications (i.e., brimonidine taken twice daily, carteolol taken once daily, lantanoprost taken once daily, simbrinza taken twice daily, or cosopt taken twice daily). The present study was approved by the Institutional Review Board of Chang Gung Memorial Hospital (IRB No.: 201802364B0C601) and was conducted in accordance with the Declaration of Helsinki.

The exclusion criteria were as follows: having an intraocular disease other than glaucoma or cataract or a history of intraocular surgery, taking an antiglaucoma medication before the present study, or having unreliable VF results (false-positive errors > 15% or fixation loss > 25%) or OCT angiography images (signal strength index < 50%). Furthermore, participants who changed their topical medication or received laser treatment during the study period were excluded.

### 2.2. Ocular Examination Data Collection

All patients underwent detailed ophthalmologic examinations that included a visual acuity assessment, pneumatic tonometry, gonioscopy, slit-lamp bio-microscopy, and OCT angiography measurements (Optovue, Fremont, CA, USA) before treatment and at 3 and 6 months after topical medication treatment. Standard automated perimetry (standard Swedish interactive threshold algorithm 30–2 test; Humphrey Field Analyzer II, Carl Zeiss Meditec, Dublin, CA, USA) was performed within 6 months of enrollment. On the basis of the Hodapp–Parrish–Anderson criteria and using the value of mean deviation (MD), the severity of glaucomatous functional damage was classified as early-stage damage (MD ≥ −6 dB), moderate-stage damage (−12 dB ≤ MD < −6 dB), or severe-stage damage (MD < −12 dB) [16].

### 2.3. OCT Angiography Measurement

For all participants, OCTA measurements (Optovue, Fremont, CA, USA) were taken at baseline and at 3 and 6 months after topical medication treatment. Motion artifacts were removed using an orthogonal registration algorithm. At the macula, the superficial and deep VD were measured in the 3.0 × 3.0 mm^2^ parafoveal region and 6.0 × 6.0 mm^2^ perifoveal region. The superficial retinal layer was segmented from the inner limiting membrane (ILM) to the inner plexiform layer (IPL); the deep retinal layer was segmented from the IPL to the outer plexiform layer. At the peripapillary region, the VD of radial peripapillary capillaries (RPCs) was determined by applying the Angio Disc protocol and measuring the region that was defined as a 750 µm-wide elliptical annulus extending from the optic disc boundary. The boundary of the RPCs extended from the ILM to the posterior retinal nerve fiber layer (RNFL).

### 2.4. Statistical Methods

The baseline demographic data of the case and control groups were compared by performing a chi-squared test and an independent samples t-test for categorical and continuous data, respectively. The correlations of baseline IOP with OCT angiography and VF parameters were investigated using Spearman’s correlation test. The quantitative OCT and OCT angiography parameters at multiple time points were assessed using a linear mixed model that was adjusted for inter-eye correlation for both the case and control groups. Eyes that experienced an IOP reduction of at least 20% from baseline were included for a subgroup analysis in which generalized estimating equations were applied. The correlations between IOP changes and OCT and OCT angiography parameter changes were determined through Spearman’s correlation test. The collected data were analyzed using SPSS Version 26.0 (IBM, Armonk, NY, USA). Statistical significance was defined with a two-tailed *p* value of < 0.05.

## 3. Results

In total, 52 eyes from 36 participants were included in the present study. Topical IOP-lowering medications were required for 33 eyes (10 POAG, 11 OHT, 12 NTG) of 17 participants. The remaining 19 healthy eyes of 19 participants for which medication was not required were used as controls.

The mean age of the 36 participants was 55.3 ± 15.9 years (25 to 77 years old). Eighteen (50%) of them were male, and the remaining 18 (50%) were female. In the case group, IOP decreased from the baseline mean of 20.4 ± 0.8 mmHg to 15.7 ± 0.5 mmHg (first visit; *p* < 0.001) and to 16.1 ± 0.5 mmHg (second visit; *p* < 0.001) after treatment. An IOP reduction of at least 20% was observed in 15 (45%) and 12 (36%) of the 33 case eyes during first and second visits, respectively. In the control group, no significant IOP changes occurred between the baseline and the first and second visits.

Table 1 presents a comparison of the baseline OCT angiography parameters of the case and control groups. Relative to the control group, the case group had a significantly higher IOP and a significantly lower MD. The case group had an average MD of −5.432 dB indicating that most of the participants in this group had early glaucoma [16]. The pattern standard deviation of the case group was also higher than that of the control group, although the difference between the two groups was nonsignificant. The baseline OCT angiography parameters of the two groups did not differ significantly after adjusting for baseline IOP. Table 2 presents the changes in OCT angiography parameters between the baseline and follow-up visits for the case and control groups, revealing that for both the case and control groups, their OCT angiography parameters did not change significantly between the baseline and first or second visit.

Table 3 presents the correlation of baseline IOP with OCT angiography parameters for both the case and control groups. Among the participants who experienced a 20% IOP reduction between the baseline and first visit, baseline IOP was negatively correlated with a deep perifoveal VD (correlation coefficient (r) = −0.551; *p* = 0.011).

Table 4 presents the correlations of the IOP changes with OCT angiography parameter changes for both the case and control groups. Among the participants who experienced a 20% IOP reduction between the baseline and second visit, a significant positive correlation between the change in their IOP and the change in their RNFL (r = 0.704; *p* = 0.011) was observed. No similar significant correlations were observed in the control group. 

Table 5 presents the results of the subgroup analysis of the eyes that experienced a 20% IOP reduction. The mean reduction in IOP was 34.1% between the baseline and first visit and 35.4% between the baseline and the second visit. The deep perifoveal VD of these eyes significantly decreased between the baseline and first visit (*p* = 0.030) but not between the baseline and second visit (*p* = 0.432). A nonsignificant reduction in their RNFL occurred between the baseline and first visit (*p* = 0.063). The subgroup analysis did not reveal any significant differences for the other parameters.

## 4. Discussion

In the present prospective study, the effect of medical IOP reduction on the ocular microvasculature of the macula, optic nerve head, and peripapillary region was determined using OCT angiography in treatment-naïve patients. Although there was no significant change in most OCT angiography parameters during the follow-up interval, our results revealed that the deep vessel density decreased initially and returned at 6 months follow-up in the subgroup analysis of cases with 20% IOP reduction, which was correlated with their baseline IOP and the IOP changes.

Optical coherence tomography (OCT) is a non-invasive method of measuring the backscattered light to reconstruct the depth profile, which is applied in glaucoma for diagnosis and detection of progression by evaluating the retinal structures, such as the retinal nerve fiber layer (RNFL), optic nerve head and ganglion cell complex (GCC) [17,18]. Yu et al. report that progressive RNFL thinning is predictive of subsequent functional decline and could be used to evaluate the effect of glaucoma treatment [19]. However, our study demonstrated that the RNFL thickness significantly decreased despite the high IOP being largely controlled by topical medication at 6 months follow-up, which indicated that IOP might not be the only factor to consider when assessing the control effect of glaucoma progression.

OCT angiography uses the movement of red blood cells as a contrast to differentiate vascular networks from static tissues [20]. Changes in the microvasculature can be observed by OCT angiography which might be correlated to glaucoma. Decreased vessel density has been reported in some studies with different performances corresponding to different glaucoma severities [3,4]; however, the causality between vessel density and the structural changes remains undetermined. 

The standard treatment for patients with glaucoma is IOP reduction by either topical medication or surgical treatment [5,6]. However, the effects of IOP reduction on microvascular change in patients with glaucoma are inconsistent. Some studies have presented an improvement in OCT angiography parameters after IOP lowering treatment [7,8,9,10,11], whereas others have not [12,13,14]. Moreover, even though medical IOP reduction is the most common initial choice for treating glaucoma, only a few studies have focused on the effect of it [7,8,15]. A cross-sectional retrospective study by Lin et al. shows that for NTG participants the different topical medications, including dorzolamide, carteolol and brimonidine, have different impacts on peripapillary superficial retinal VD and RNFL [8]. A prospective study by Liu et al. reports the lowering of IOP around 26% by Latanoprost is significantly correlated with an increase in VD in the ONH and RPC [7]. Another prospective study by Kurysheva et al. reports a significant decrease in VD in the ONH in patients with early-stage glaucoma under Tafluprost or Tafluprost/Timolol, based on weekly follow-up [12].

Our study showed that baseline IOP was negatively correlated with deep perifoveal VD in cases with 20% IOP reduction between baseline and the 3-month follow-up, which indicated that the higher baseline IOP might have influenced the deep perifoveal region and that vessel insufficiency would be observed in the short term. After the IOP was reduced by topical medication, the impact on VD over the deep perifoveal region was compromised at a 3-month follow-up. Most likely, these patients who were sensitive to IOP-lowering topical medication suffered a rapid plunge in IOP, which increased the lability of the deep parafoveal tissue and led to the temporary reduction in vessel density. Choi M et al. found that perifoveal microcirculation was vulnerable at high IOP, especially in the deep vascular complex (DVC) [21]. The capillaries of the DVC are the terminal capillary anastomotic circulation supplied from the descending precapillary arteriole from the superficial vascular complex, resulting in higher flow resistance. Furthermore, the laminar planes of the DVC without vertical exit of the arterial branch supply might explain why this area is much more susceptible to fluctuations of IOP [21,22]. The novelty of our study was that it demonstrated that deep capillary VD of the macula is correlated to baseline IOP, decreased deep-layer VD and subsequent RNFL thinning during medical treatment for early glaucoma with IOP control.

The strength of the present study is its status as a prospective study investigating the connection between topical IOP lowering medications in glaucoma patients and vessel density parameters gleaned from OCTA. The two follow-ups at 3 and 6 months reveal the short-term changes in OCTA during the IOP lowering process in early glaucoma. 

The major limitation of this study is the small and heterogenous sample size with short follow-up period after consent was withdrawn during the COVID-19 pandemic. This meant that further, regular VF examinations could not be conducted during the study period. Additionally, the largely homogenous group of glaucoma participants received a variety of topical IOP-lowering medications, which needs to be elucidated in further study.

## 5. Conclusions

In this prospective study, the preliminary result demonstrated that macular deep capillary VD was compromised and negatively correlated to baseline IOP in the 20% IOP reduction subgroup. Furthermore, for early glaucoma, the macula deep VD was vulnerable and decreased at 3 months, whereas RNFL thickness change was positively correlated to IOP change at 6 months follow-up. OCTA parameters provided in vivo monitoring information during medical treatment for glaucoma.

## Figures and Tables

**Table 1 diagnostics-12-02174-t001:** Comparison of the baseline parameters between cases and controls.

	Cases Mean (SE)	Controls Mean (SE)	Adjusted Mean Difference (95% CI) *	*p* *
Baseline IOP (mmHg)	20.423 ± 0.806	15.658 ± 0.578	4.765 (2.713 to 6.816)	**<0.001**
RPC VD (%)	48.670 ± 1.211	47.615 ± 1.292	1.055 (–2.470 to 4.581)	0.54
Superficial parafoveal VD (%)	44.362 ± 1.196	44.705 ± 1.396	–0.343 (–4.095 to 3.410)	0.852
Deep parafoveal VD (%)	50.783 ± 0.986	51.904 ± 1.070	–1.121 (–4.235 to 1.994)	0.47
Superficial perifoveal VD (%)	45.312 ± 1.287	45.880 ± 1.191	–0.568 (–4.285 to 3.149)	0.758
Deep perifoveal VD (%)	45.273 ± 1.358	46.439 ± 1.577	–1.166 (–5.299 to 2.968)	0.56
RNFL (µm)	93.202 ± 2.244	91.858 ± 2.472	1.343 (–5.020 to 7.707)	0.656
GCC (µm)	89.850 ± 2.058	89.143 ± 1.814	0.707 (–5.055 to 6.470)	0.802
VF MD (dB)	–5.432 ± 1.166	–1.484 ± 0.897	–3.948 (–7.237 to –0.658)	**0.02**
VF PSD (dB)	5.404 ± 0.766	3.246 ± 0.698	2.158 (–0.068 to 4.384)	0.057

Continuous data are presented as mean ± standard error. Abbreviations: CI: confidence interval; GCC: ganglion cell complex; IOP: intraocular pressure; RNFL: retinal nerve fiber layer; RPC: radial peripapillary capillary network; SE: standard error; VD: vessel density; VF MD: visual field mean deviation; VF PSD: visual field pattern standard deviation. Bold values indicate *p* < 0.05. * Adjusted for baseline IOP.

**Table 2 diagnostics-12-02174-t002:** Comparison of the optical coherence tomography angiography parameters at baseline and follow-up visits in the cases and controls.

	Cases				Controls			
	**Baseline Mean (SE)**	**Visit 1 Mean (SE)**	**Adjusted Mean Difference (95% CI) ***	***p* ***	**Baseline Mean (SE)**	**Visit 1 Mean (SE)**	**Adjusted Mean Difference (95% CI) ***	***p* ***
IOP (mmHg)	20.405 (0.809)	15.666 (0.497)	–4.739 (–7.001 to –2.477)	**<0.001**	15.658 (0.578)	15.568 (0.454)	–0.89 (–1.284 to 1.106)	0.913
RPC VD (%)	48.192 (1.341)	48.007 (1.632)	–0.185 (–1.734 to 1.364)	0.929	48.095 (1.224)	48.247 (1.244)	0.153 (–1.163 to 1.468)	1
Superficial parafoveal VD (%)	43.881 (1.249)	45.027 (1.201)	1.145 (–0.352 to 2.643)	0.177	45.337 (1.289)	44.774 (1.268)	–0.563 (–2.394 to 1.268)	1
Deep parafoveal VD (%)	50.859 (0.948)	51.453 (0.884)	0.594 (–1.360 to 2.548)	1	51.526 (0.920)	49.621 (1.086)	–1.905 (–3.958 to 0.148)	0.072
Superficial perifoveal VD (%)	44.967 (1.286)	45.124 (1.313)	0.158 (–1.209 to 1.524)	1	46.305 (1.110)	46.321 (1.113)	0.016 (–1.437 to 1.469)	1
Deep perifoveal VD (%)	44.773 (1.514)	44.515 (1.490)	–0.258 (–2.255 to 1.740)	1	47.063 (1.545)	45.953 (1.442)	–1.111 (–3.470 to 1.249)	0.551
RNFL (µm)	92.466 (2.619)	91.799 (2.607)	–0.667 (–3.303 to 1.970)	1	93.105 (2.358)	92.684 (2.439)	–0.421 (–1.847 to 1.004)	1
GCC (µm)	89.259 (2.104)	89.168 (2.117)	–0.091 (–0.691 to 0.509)	1	90.158 (1.833)	91.000 (1.691)	0.842 (–1.782 to 3.466)	0.605
	**Baseline Mean (SE)**	**Visit 2 Mean (SE)**	**Adjusted Mean Difference (95% CI) ***	***p* ***	**Baseline Mean (SE)**	**Visit 2 Mean (SE)**	**Adjusted Mean Difference (95% CI) ***	***p* ***
IOP (mmHg)	20.405 (0.809)	16.066 (0.495)	–4.339 (–6.421 to –2.258)	**<0.001**	15.658 (0.578)	15.179 (0.557)	–0.479 (–1.897 to –0.940)	0.913
RPC VD (%)	48.192 (1.341)	47.562 (1.543)	–0.630 (–2.128 to 0.867)	0.929	48.095 (1.224)	48.137 (1.166)	0.042 (–1.265 to 1.349)	1
Superficial parafoveal VD (%)	43.881 (1.249)	44.630 (1.380)	0.748 (–1.337 to 2.834)	0.835	45.337 (1.289)	44.326 (1.354)	–1.011 (–3.766 to 1.745)	1
Deep parafoveal VD (%)	50.859 (0.948)	50.704 (1.290)	–0.155 (–2.812 to 2.503)	1	51.526 (0.920)	49.732 (1.107)	–1.795 (–3.757 to 0.167)	0.076
Superficial perifoveal VD (%)	44.967 (1.286)	44.897 (1.586)	–0.070 (–1.346 to 1.206)	1	46.305 (1.110)	46.995 (0.759)	0.689 (–1.465 to 2.844)	1
Deep perifoveal VD (%)	44.773 (1.514)	43.824 (1.660)	–0.948 (–4.184 to 2.287)	1	47.063 (1.545)	45.258 (1.122)	–1.805 (0.278 to 0.820)	0.278
RNFL (µm)	92.466 (2.619)	92.345 (2.869)	–0.121 (–2.312 to 2.070)	1	93.105 (2.358)	93.158 (2.393)	0.053 (–1.277 to 1.383)	1
GCC (µm)	89.259 (2.104)	88.956 (2.070)	–0.303 (–1.626 to 1.020)	1	90.158 (1.833)	90.632 (1.793)	0.474 (–0.627 to 1.574)	0.653

* Linear mixed model: adjusted for inter-eye correlation. Abbreviations: CI: confidence interval; GCC: ganglion cell complex; IOP: intraocular pressure; RNFL: retinal nerve fiber layer; RPC: radial peripapillary capillary network; SE: standard error; VD: vessel density; VF MD: visual field mean deviation; VF PSD: visual field pattern standard deviation. Bold values indicate *p* < 0.05.

**Table 3 diagnostics-12-02174-t003:** Correlation of baseline intraocular pressure with optical coherence tomography angiography parameters.

	Cases (ALL) Correlation Coefficient (*p*)	Cases (20% IOP Reduction from Baseline to Visit 1) Correlation Coefficient (*p*)	Cases (20% IOP Reduction from Baseline to Visit 2) Correlation Coefficient (*p*)	Controls Correlation Coefficient (*p*)
RPC VD (%)	–0.118 (0.513)	–0.152 (0.589)	0.006 (0.984)	0.275 (0.255)
Superficial parafoveal VD (%)	–0.203 (0.257)	–0.420 (0.119)	–0.147 (0.649)	0.323 (0.178)
Deep parafoveal VD (%)	0.018 (0.920)	–0.005 (0.985)	0.328 (0.298)	0.188 (0.442)
Superficial perifoveal VD (%)	–0.061 (0.736)	–0.407 (0.132)	–0.342 (0.276)	0.240 (0.322)
Deep perifoveal VD (%)	–0.233 (0.192)	–0.551 (**0.033**)	–0.230 (0.472)	–0.197 (0.419)
RNFL (µm)	–0.268 (0.131)	–0.294 (0.287)	–0.224 (0.484)	0.125 (0.611)
GCC (µm)	–0.213 (0.235)	–0.389 (0.152)	–0.341 (0.278)	–0.257 (0.287)

Abbreviations: GCC: ganglion cell complex; IOP: intraocular pressure; RNFL: retinal nerve fiber layer; RPC: radial peripapillary capillary network; VD: vessel density. Bold values indicate *p* < 0.05.

**Table 4 diagnostics-12-02174-t004:** Correlation of changes in intraocular pressure with changes in optical coherence tomography angiography parameters.

	Baseline to Visit 1			Baseline to Visit 2		
	Cases (ALL) Correlation Coefficient (*p*)	Cases (20% IOP Reduction) Correlation Coefficient (*p*)	Controls Correlation Coefficient (*p*)	Cases (ALL) Correlation Coefficient (*p*)	Cases (20% IOP Reduction) Correlation Coefficient (*p*)	Controls Correlation Coefficient (*p*)
Change in RPC VD	–0.237 (0.185)	0.465 (0.081)	0.090 (0.715)	–0.148 (0.412)	0.503 (0.095)	–0.087 (0.723)
Change in superficial parafoveal VD	–0.016 (0.930)	–0.063 (0.825)	–0.363 (0.127)	–0.039 (0.828)	–0.385 (0.217)	0.025 (0.918)
Change in deep parafoveal VD	–0.206 (0.250)	0.399 (0.141)	–0.182 (0.455)	0.108 (0.551)	–0.067 (0.837)	0.040 (0.870)
Change in superficial perifoveal VD	–0.042 (0.817)	–0.023 (0.934)	–0.263 (0.276)	0.033 (0.855)	–0.455 (0.138)	0.285 (0.238)
Change in deep perifoveal VD	–0.075 (0.679)	–0.201 (0.473)	–0.247 (0.308)	–0.090 (0.620)	0.119 (0.713)	0.054 (0.827)
Change in RNFL	0.256 (0.150)	0.201 (0.474)	0.103 (0.676)	–0.021 (0.907)	0.704 (**0.011**)	0.410 (0.081)
Change in GCC	0.030 (0.867)	0.004 (0.989)	0.138 (0.572)	–0.255 (0.152)	0.450 (0.142)	–0.232 (0.338)

Abbreviations: GCC: ganglion cell complex; IOP: intraocular pressure; RNFL: retinal nerve fiber layer; RPC: radial peripapillary capillary network; VD: vessel density. Bold values indicate *p* < 0.05.

**Table 5 diagnostics-12-02174-t005:** Comparison of the OCTA parameters at baseline and follow-up visits in cases which demonstrated at least 20% IOP reduction.

		Visit 1 vs. Baseline (15 Eyes)			Visit 2 vs. Baseline (12 Eyes)		
	Baseline Mean (SE)	Visit 1 Mean (SE)	Adjusted Mean Difference (95% CI) *	*p* Value *	Visit 2 Mean (SE)	Adjusted Mean Difference (95% CI) *	*p* Value *
RPC VD (%)	47.782 (2.041)	47.132 (2.264)	–0.65 (–1.531 to 0.231)	0.148	47.216 (2.514)	–0.567 (–1.498 to 0.365)	0.233
Superficial parafoveal VD (%)	45.35 (2.735)	45.767 (1.967)	0.417 (–1.73 to 2.563)	0.704	46.533 (2.399)	1.183 (–0.455 to 2.821)	0.157
Deep parafoveal VD (%)	52.916 (2.063)	50.133 (2.354)	–2.783 (–5.296 to –0.27)	**0.030**	51.249 (1.480)	–1.667 (0.12 to –3.766)	0.432
Superficial perifoveal VD (%)	46.062 (2.926)	45.095 (2.167)	–0.967 (–3.409 to 1.476)	0.438	46.262 (2.300)	0.2 (–1.575 to 1.975)	0.825
Deep perifoveal VD (%)	45.546 (3.345)	45.396 (3.374)	–0.15 (–5.083 to 4.783)	0.952	46.229 (1.971)	0.683 (–4.818 to 6.185)	0.808
RNFL (µm)	88.912 (7.016)	88.078 (7.044)	–0.833 (–1.711 to 0.045)	0.063	88.078 (7.537)	–0.833 (–2.569 to 0.902)	0.347
GCC (µm)	87.33 (4.521)	87.497 (4.416)	0.167 (–0.648 to 0.982)	0.688	87.663 (4.488)	0.333 (–0.338 to 1.005)	0.330

* Generalized estimating equation: adjusted for inter-eye correlation. Abbreviations: CI: confidence interval; GCC: ganglion cell complex; IOP: intraocular pressure; RNFL: retinal nerve fiber layer; RPC: radial peripapillary capillary network; SE: standard error; VD: vessel density. Bold values indicate *p* < 0.05.

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
