# Peer review of "Association of Intraocular Pressure and Optical Coherence Tomography Angiography Parameters in Early Glaucoma Treatment"

_diagnostics, 2022, doi:10.3390/diagnostics12092174_

Round 1

Reviewer 1 Report

some corrections in the text are needed:

line 13-14 - instead of biomicroscope - it should be biomicroscopy

lines 22-23, 195, 197, 198-199, 207-208 and 215-216 - pl. check. Meaning not clear.

Conclusion - pl. check the meaning, not very clear.

line 250 - abbreviation of journals should be as per recommendations. Jama should be JAMA.

line 287 - pages are missing. pl. provide.

Author Response

Response to Reviewer 1 Comments

Point 1: line 13-14 - instead of biomicroscope - it should be biomicroscopy

Response 1: Thanks for the suggestion. We have corrected it to “bio-microscopy”. The revision appears in line 13.

Point 2: lines 22-23, 195, 197, 198-199, 207-208 and 215-216 - pl. check. Meaning not clear.

Response 2: Thank you for this informative suggestion. We listed the explanations below.

lines 22-23 Original -For the subgroup with an IOP reduction of >20%, the deep macula VD significantly decreased and was negatively correlated with baseline IOP at 3-months follow-up.

Answer: For the subgroup with an IOP reduction of >20%, the deep macula VD was negatively correlated with baseline IOP and had decreased at 3-months follow-up. Additionally, our results (Table 4) suggested that under optimal IOP control, RNFL thinning might have worsened during disease monitoring. Hence, in our prospective study, OCTA provided useful information about early glaucoma cases. To clarify these points in our abstract, we made some revisions and added some points to make our meaning clear. The revisions appear in lines 24-27.

line 195 Original -Lin et al. cross-sectional retrospective study showed that the different topical IOP lowering medication has different alterations at peripapillary superficial retinal VD and RNFL [8].

Answer: In the retrospective study by Lin et al., the topical medications, including dorzolamide, carteolol and brimonidine for normal tension glaucoma, yielded varying results. We have added this point in the revision, which appears in lines 198-200.

line 197 Original -Liu et al. prospective studies reported the lowering of IOP by Latanoprost was significantly correlated to the increase of VD in ONH and RPC [7].

Answer: In the first paragraph of our introduction section, we introduce the abbreviations for optic nerve head (ONH) and radial peripapillary capillaries (RPC). In Liu’s prospective study with higher IOP reduction of around 26% by latanoprost, a prostaglandin analogue, IOP reduction was found to be correlated to an increase in VD in the optic nerve head (ONH) and radial peripapillary capillaries (RPC). The revision appears in line 201.

lines 198-199 Original- Another prospective studies of Kurysheva et al. presented that significant decrease of VD in the ONH in patients with early stage of glaucoma under Tafluprost or Tafluprost/Timolol for one week [12].

Answer: This prospective study showed that the OCTA parameters changed during short-term follow-up based on a weekly interval. The revision appears in lines 203-205.

lines 207-208 Original -In the subgroup analysis of patients with 20% IOP reduction, the deep parafoveal VD significantly decreased was observed at 3-month follow-up.

Answer: This part is a repeat of the previous sentence (lines 208-209), and therefore we have deleted this sentence.

line 215-216 “What’s more, the laminar structure of the DVC without vertically exit of arterial branches which might explain that why this area are much susceptible to the fluctuation of IOP and vessel insufficiency [21,22].”

Answer: Although parapapillary vessel density change after treatment had been evaluated in the studies by Lin et al., Liu et al. and Kurysheva et al. (references 7, 8 and 12), reports regarding macular superficial or deep vessel density are lacking. Recently there has been some literature regarding retinal interface disease with glaucoma. With regard to the long-term care of glaucoma patients, this issue regarding the change of microcirculation is an unmet need. (Ref: Vitreomacular Interface Abnormalities and Glaucoma in an Elderly Population (The MONTRACHET Study; Julie Blanc, Alassane Seydou, Inès Ben Ghezala, Clémence Deschasse, Cyril Meillon, Alain M Bron, Christine Binquet, Catherine Creuzot-Garcher. Invest Ophthalmol Vis Sci. 2019 May 1;60(6):1996-2002.) We have corrected the original sentences to clarify and at the end of this paragraph we have added “The novelty of our study was that it demonstrated that deep capillary VD of the macula is correlated to baseline IOP, decreased deep layer VD and subsequent RNFL thinning during medical treatment for early glaucoma with IOP control.“ The revision appears in lines 219-222.

Point 3: Conclusion - pl. check the meaning, not very clear.

Response 3: Thank you for the comment. We have rewritten the conclusion as per your suggestion. In this prospective study, the preliminary result demonstrated that macular deep capillary VD was compromised and negatively correlated to baseline IOP in a 20% IOP reduction subgroup. Furthermore, for early glaucoma, the macula deep VD is vulnerable and decreased at 3 months and RNFL thickness change is positively correlated to IOP change at 6 months follow-up. OCTA parameters provide in-vivo monitoring information during the medical treatment for glaucoma. The revisions appear in lines 234-239.

Point 4: abbreviation of journals should be as per recommendations. Jama should be JAMA.

Response 4: We have corrected this to “JAMA”. The revisions appear in line 260.

Point 5: pages are missing. pl. provide.

Response 5: Thanks for the suggestion. The revision appears in line 297.

Reviewer 2 Report

The article is suitable for publication in the submitted form. No comments or corrections to be added.

Reviewer 3 Report

The paper entitled “Association of intraocular pressure and optical coherence tomography angiography parameters in early glaucoma treatment” is a study based on the effects of medical intraocular pressure reduction on vascular structures assessed with optical coherence tomography (OCT) angiography in early glaucoma. The manuscript is of potential clinical interest considering the importance of OCT in the management of glaucoma. The paper, however, has several important flaws.

The study plan for this prospective study is quite limiting. Goldmann applanation tonometry is the gold standard in diagnosing and managing glaucoma. It is not clear why the author selected pneumatic tonometry to assess glaucoma patients. This type of tonometry is useful in screening but surely not appropriate in a clinical glaucoma setting.

The authors included OHT patients in this study who, by definition do not have structural and/or functional glaucomatous damage. Normal tension glaucoma patients were also included. Considering that the topic of the study is to assess vascular defects related to IOP changes in glaucoma patients, the inclusion of these patients may probably not be best suited to assess this hypothesis. A cohort of homogenous glaucoma patients using medical therapy to lower IOP would have been preferred to examine the true influence of lowering IOP in glaucomatous patients.

The authors stated in the results section, that a total of 19 healthy eyes did not require medication and were used as controls. Considering that inclusion criteria were based on a diagnosis of glaucoma, ocular hypertension, and normal-tension glaucoma, it seems inappropriate and biased to define patients with these conditions that did not use medical therapy as “controls” in this type of study.

The fact that this is a prospective study is great, however, the short follow-up, the use of both eyes in the same patient, heterogeneous cohort, and small numbers are rather limiting and not sufficient to make conclusive remarks, which need to be toned down in the conclusion section. These important limitations need to be addressed if possible or listed in the discussion section as major limits. The study results should be presented as only very preliminary and introductory, which could be useful for future un depth studies regarding this issue.

The potential effects of IOP on the vascular system are quite interesting. Further explanations, possible physiopathological mechanisms, and hypotheses should be added in the Discussion section considering that this is the main topic of the paper.

Author Response

Response to Reviewer 2 Comments

Point 1: The paper entitled “Association of intraocular pressure and optical coherence tomography angiography parameters in early glaucoma treatment” is a study based on the effects of medical intraocular pressure reduction on vascular structures assessed with optical coherence tomography (OCT) angiography in early glaucoma. The manuscript is of potential clinical interest considering the importance of OCT in the management of glaucoma. The paper, however, has several important flaws.

The study plan for this prospective study is quite limiting. Goldmann applanation tonometry is the gold standard in diagnosing and managing glaucoma. It is not clear why the author selected pneumatic tonometry to assess glaucoma patients. This type of tonometry is useful in screening but surely not appropriate in a clinical glaucoma setting.

Response 1: Thank you for the comment. We are aware that Goldmann applanation is the gold standard for IOP measurement. However, as we mentioned, since our study took place during the pandemic, in the interest of our participants we consistently checked the IOP with pneumatic tonometry throughout the study period in order to avoid the use of topical anesthesia and close contact with the cornea. Furthermore, the literature has demonstrated that the accuracy is comparable to Goldmann applanation. (Ref: Jorge J, Diaz-Rey JA, Gonzalez-Meijome JM, Almeida JB, Parafita MA. Clinical performance of the Reichert AT550: a new non-contact tonometer. Ophthalmic Physiol Opt. 2002;22(6):560–4.)

Point 2: The authors included OHT patients in this study who, by definition do not have structural and/or functional glaucomatous damage. Normal tension glaucoma patients were also included. Considering that the topic of the study is to assess vascular defects related to IOP changes in glaucoma patients, the inclusion of these patients may probably not be best suited to assess this hypothesis. A cohort of homogenous glaucoma patients using medical therapy to lower IOP would have been preferred to examine the true influence of lowering IOP in glaucomatous patients.

Response 2: Thank you for the comment. As we mentioned in our discussion section, there are controversial results regarding ocular microcirculation change following topical medication. Given the diversity of severity of neuroretinal atrophy, the response to IOP reduction can be presumed to be quite complex. Furthermore, it is debatable whether the retinal vessel density and retinal nerve fiber layer thickness are an epiphenomenon or are a mutual cause resulting in glaucoma progression. Reports regarding macular superficial or deep vessel density following topical medication are lacking.

The purpose of our prospective study was to investigate the impact of microcirculation in early glaucoma with mild glaucomatous change of the optic nerve and its correlation to IOP change. Therefore, following your kind suggestion we have added the following as a limitation of our study: ”Additionally, the largely homogenous group of glaucoma participants received a variety of topical IOP-lowering medications, which needs to be elucidated in further study.” The revision appears in lines 230-231.

Point 3: The authors stated in the results section, that a total of 19 healthy eyes did not require medication and were used as controls. Considering that inclusion criteria were based on a diagnosis of glaucoma, ocular hypertension, and normal-tension glaucoma, it seems inappropriate and biased to define patients with these conditions that did not use medical therapy as “controls” in this type of study.

Response 3: Thanks for the comment. In this study, the enrolment criterion for the study group was newly diagnosed glaucoma with visual field defect or ocular hypertension. In the method for the statistical analysis we described: “The quantitative OCT and OCT angiography parameters at multiple time points were assessed using a linear mixed model that was adjusted for inter-eye correlation for both the case and control groups” (line 104). In Table 3, we analyzed with a linear mixed model and in Table 5 we analyzed with a generalized estimating equation and both results were adjusted for inter-eye correlation. Footnotes are shown for both tables.

Listed below is some literature regarding paired eye studies to evaluate the treatment effect or the visual field defect progression.

  • Selective laser trabeculoplasty for early glaucoma: analysis of success predictors and adjusted laser outcomes based on the untreated fellow eye. Chun M, Gracitelli CP, Lopes FS, Biteli LG, Ushida M, Prata TS. BMC Ophthalmol. 2016 Nov 23;16(1):206.

  • Association of Myopic Optic Disc Deformation with Visual Field Defects in Paired Eyes with Open-Angle Glaucoma: A Cross-Sectional Study. Yu Sawada, Masanori Hangai, Makoto Ishikawa , Takeshi Yoshitomi. PLoS One.2016 :29;11(8):e0161961.

  • Tutorial on Biostatistics: Linear Regression Analysis of Continuous Correlated Eye Data.Gui-Shuang Ying , Maureen G Maguire , Robert Glynn , Bernard Rosner. Ophthalmic Epidemiol. 2017 Apr;24(2):130-140.

Point 4: The fact that this is a prospective study is great, however, the short follow-up, the use of both eyes in the same patient, heterogeneous cohort, and small numbers are rather limiting and not sufficient to make conclusive remarks, which need to be toned down in the conclusion section. These important limitations need to be addressed if possible or listed in the discussion section as major limits. The study results should be presented as only very preliminary and introductory, which could be useful for future un depth studies regarding this issue.

Response 4: Thank you for the constructive comment. We have addressed these important issues in the limitations section of our study. During the pandemic, in our clinical practice, it was crucial to overcome difficulties and treat the patients, aiming to prevent vision deterioration for glaucoma patients. In the discussion section, we discussed the recent studies on topical glaucoma agents; studies by Lin et al, Liu et al, and Kurysheva et al., each with varying study periods of one week to 6 months (ref. 7,8 and 12). As we know, glaucoma is a disease spectrum, with chronic change, and there is an unmet need to explore and monitor structural change before visual function changes set in. Therefore, we have rewritten the conclusion to illustrate and strengthen the preliminary nature of our study. The revision appears in lines 234-239.

Point 5: The potential effects of IOP on the vascular system are quite interesting. Further explanations, possible physiopathological mechanisms, and hypotheses should be added in the Discussion section considering that this is the main topic of the paper.

Response 5: Thank you for the constructive comment. In the discussion section (line 219) we have cited some literature to explain how the macula deep layer VD is vulnerable to IOP fluctuation on account of its anatomy. Furthermore, it is debatable whether retinal vessel density and retinal nerve fiber layer thickness are an epiphenomenon or a mutual cause, resulting in glaucoma progression. Therefore, in the discussion section we have added: “The novelty of our study was that it demonstrated that deep capillary VD of the macula is correlated to baseline IOP, decreased deep layer VD and subsequent RNFL thinning during medical treatment for early glaucoma with IOP control.” The revision appears in lines 219-222.

Reviewer 4 Report

The authors aimed to explore the effect of medical IOP reduction and capillary vessel density by A-OCT and they found that the vessel density was influenced by IOP reductio within medical treatment.

The glaucoma scientific specialist already knows the topic of the manuscript. The main flaw of this research is the low sample size of 17 glaucoma patients.

The manuscript needs to only include one eye from each patient and to include a sample size calculation on the statistical approach.

In addition, when the sample size should be higher than actual the p value could be lower and more statistically significant differences were achieved.

The heterogeneity in the follow up is a major limitation. The authors have disclosed very well all these limitations on the discussion section, but it is exceedingly difficult to approve the continue of the publication with this several limitation

My recommendation is to improve the sample size and follow-up include these results on the manuscript and submit again in the next future.

Author Response

Response to Reviewer 3 Comments

Point 1: The authors aimed to explore the effect of medical IOP reduction and capillary vessel density by A-OCT and they found that the vessel density was influenced by IOP reductio within medical treatment. The glaucoma scientific specialist already knows the topic of the manuscript. The main flaw of this research is the low sample size of 17 glaucoma patients.

Response 1: Thanks for the comment. In our discussion section, we mentioned the updated literature for parapapillary VD change, namely Lin, Liu and Kurysheva’s studies on topical medication in glaucoma treatment. There are very few studies that prospectively demonstrate the macular VD changes in early glaucoma. We agree that the small sample size of 33 glaucoma eyes constitutes the major limitation of this study and we have acknowledged it in our discussion section. Furthermore, we have added “The novelty of our study was that it demonstrated that deep capillary VD of the macula is correlated to baseline IOP, decreased deep layer VD and subsequent RNFL thinning during medical treatment for early glaucoma with IOP control.” in the discussion section to address the novelty of this study. The revisions appear in lines 219-222.

Point 2: The manuscript needs to only include one eye from each patient and to include a sample size calculation on the statistical approach.

Response 2: Thank you for the comment. In this study, the enrolment criterion for the study group was newly diagnosed glaucoma with visual field defect or ocular hypertension. In the method for the statistical analysis, we described: “The quantitative OCT and OCT angiography parameters at multiple time points were assessed using a linear mixed model that was adjusted for inter-eye correlation for both the case and control groups”. In Table 3, we analyzed with a linear mixed model and in Table 5 we analyzed with a generalized estimating equation and both results were adjusted for inter-eye correlation. Footnotes are shown for both tables.

Listed below is some literature regarding paired eye studies to evaluate the treatment effect or the visual field defect progression.

  • Selective laser trabeculoplasty for early glaucoma: analysis of success predictors and adjusted laser outcomes based on the untreated fellow eye. Chun M, Gracitelli CP, Lopes FS, Biteli LG, Ushida M, Prata TS. BMC Ophthalmol. 2016 Nov 23;16(1):206.

  • Association of Myopic Optic Disc Deformation with Visual Field Defects in Paired Eyes with Open-Angle Glaucoma: A Cross-Sectional Study. Yu Sawada, Masanori Hangai, Makoto Ishikawa , Takeshi Yoshitomi. PLoS one. 2016 :29;11(8):e0161961.

  • Tutorial on Biostatistics: Linear Regression Analysis of Continuous Correlated Eye Data.Gui-Shuang Ying , Maureen G Maguire , Robert Glynn , Bernard Rosner. Ophthalmic Epidemiol. 2017 Apr;24(2):130-140.

Point 3: In addition, when the sample size should be higher than actual the p value could be lower and more statistically significant differences were achieved.

Response 3: Thank you for the suggestion, we will continue to conduct the study and obtain long-term follow-up results in the near future. In clinical practice, we regularly check IOP and examine visual field function. The ocular microcirculation changes after we initiate medication and therefore it is important to monitor the structural and functional progress. Based on the data, there is a significant correlation between OCTA parameters and IOP change in early glaucoma during treatment. In our discussion section (line 219) we have cited some literature to explain how the macula deep layer VD is vulnerable to IOP fluctuation on account of its anatomy. Our preliminary result also supports the vascular theory of glaucoma mechanism. (Ref: Flammer J, Orgul S, Costa VP, et al. The impact of ocular blood flow in glaucoma. Prog Retin Eye Res. 2002;21:359–93.)

Point 4: The heterogeneity in the follow up is a major limitation. The authors have disclosed very well all these limitations on the discussion section, but it is exceedingly difficult to approve the continue of the publication with this several limitations. My recommendation is to improve the sample size and follow-up include these results on the manuscript and submit again in the next future.

Response 4: Thank you for the comment. We have listed these important issues as a major limitation of our study. During the pandemic, in our clinical practice, it was crucial to overcome difficulties and treat the patients, aiming to prevent vision deterioration for glaucoma patients. In the discussion section, we discussed the recent studies on topical glaucoma agents; studies by Lin et al, Liu et al, and Kurysheva et al., each with varying study periods of one week to 6 months (ref. 7,8 and 12). As we know, glaucoma is a disease spectrum, with chronic change, and there is an unmet need to explore and monitor structural change before visual function changes set in. Therefore, we have rewritten the conclusion to illustrate and strengthen the preliminary nature of our study. The revision appears in lines 234-239. Based on our preliminary results, we suggest prospective studies with large, homogenous, sample sizes and enrolling glaucoma patients receiving different medications should be conducted in the near future.

Round 2

Reviewer 3 Report

The authors have addressed the issues in a satisfactory manner.

Reviewer 4 Report

Despite the fact that the authors answer all the question, the major flaws of the manuscript have not solved. 

The inclusion in the limitation section is not enough to improve the quality of the paper. 

There are some unsolved issues explain on the first round of revision that the authors could no solve in 10 days

Thank you very much for your time and effort

My decision continue on reject

Thanks to the journal to trust in me to prepare this revision